# WorldModelBench: Judging Video Generation Models As World Models

**Dacheng Li**[1*]  **Yunhao Fang**[2*]  **Yukang Chen**[3]  **Shuo Yang**[1]  **Shiyi Cao**[1]

**Justin Wong**[1]  **Michael Luo**[1]  **Xiaolong Wang**[2,3]  **Hongxu Yin**[3]

**Joseph E. Gonzalez**[1]  **Ion Stoica**[1]  **Song Han**[3,4]  **Yao Lu**[3]

[1]UC Berkeley  [2]UC San Diego  [3]NVIDIA  [4]MIT

## Abstract

Video generation models have rapidly progressed, positioning themselves as video world models capable of supporting decision-making applications like robotics and autonomous driving. However, current benchmarks fail to rigorously evaluate these claims, focusing only on general video quality, ignoring important factors to world models such as physics adherence. To bridge this gap, we propose WorldModel-Bench, a benchmark designed to evaluate the world modeling capabilities of video generation models in application-driven domains. WorldModelBench offers two key advantages: (1) **Against to nuanced world modeling violations**: By incorporating instruction-following and physics-adherence dimensions, WorldModel-Bench detects subtle violations, such as irregular changes in object size that breach the mass conservation law—issues overlooked by prior benchmarks. (2) **Aligned with large-scale human preferences**: We crowd-source 67K human labels to accurately measure 14 frontier models. Using our high-quality human labels, we further fine-tune an accurate judger to automate the evaluation procedure, achieving 9.9% lower error in predicting world modeling violations than GPT-4o with 2B parameters. In addition, we demonstrate that training to align human annotations by maximizing the rewards from the judger noticeably improve the world modeling capability. The dataset is hosted in HuggingFace at `https://huggingface.co/datasets/Efficient-Large-Model/worldmodelbench`. The code to run evaluation is available at `https://github.com/WorldModelBench-Team/WorldModelBench`.

## 1 Introduction

Video generation models have achieved remarkable success in creating high-fidelity and realistic videos [24, 8, 44, 57, 13, 52, 62, 42, 29, 18]. Beyond generating visually compelling content, these models are increasingly seen as potential **video world models**. Video world models simulate feasible future frames based on given text and image instruction [31, 42, 1]. These future frames obey real-world dynamics and unlock grounded planning on decision-making tasks such as robotics, autonomous driving, and human body prediction [6, 7, 1, 63, 9, 19, 10].

Despite the potential, the ability of video generation models to act as reliable world models remains speculative. Existing benchmarks primarily evaluate on general video quality such as temporal

---

*Equal contribution. Part of the work was done while Dacheng Li and Yunhao Fang were interns at NVIDIA. Correspondence to: `dacheng177@berkeley.edu`, `yuf026@ucsd.edu`, `songhan@mit.edu`, `jasonlu@nvidia.com`.

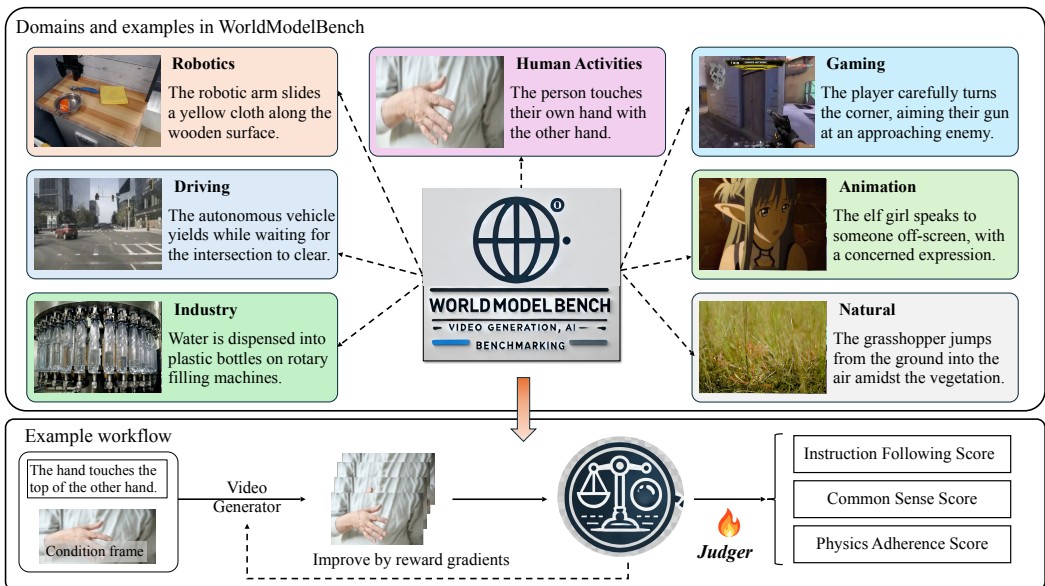

Figure 2: **Overview of WorldModelBench**. WorldModelBench **judges** the **world modeling** capability of video generation models across diverse **application-driven** domains. On WorldModelBench, a model generates a video based on text and optionally image conditions and is scored along **common-sense**, **instruction following**, and **physics adherence** dimensions. We collect 67K **human labels** to evaluate 14 frontier models. WorldModelBench is paired with a fine-tuned judger, providing fine-grained feedback for models, and training to aligns its reward improves world modeling capabilities.

consistency and aesthetic coherence [26, 36, 54]. While these measures are necessary for video world models, they are inadequate. Importantly, they do not capture real-world dynamics, e.g. adhere to basic real-world physics (Figure 1). While efforts like VideoPhy [4] introduce physics-based evaluations, their focus on interactions between daily objects overlooks broader application-driven scenarios.

To address the gap, we introduce WorldModelBench to judge the world modeling capability of video generation models. WorldModelBench consists of 350 image and text condition pairs, ranging over 7 application driven domains, 56 diverse subdomains, and provides support for both text-to-video (T2V) and image-to-video (I2V) models. In addition to being a comprehensive benchmark, WorldModelBench features two **unique** advantages.

*Firstly*, WorldModelBench detects nuanced world modeling violations that are overlooked by previous benchmarks. WorldModelBench maintains a minimal evaluation on general video quality (frame-wise and temporal quality), and focuses to introduce two dimensions specifically for world modeling: instruction following and physics adherence. It further provides fine-grained categories for these two dimensions to capture nuances: instruction following dimension is broken down into four levels and physics adherence are listed into five common violations (§ 3.1). By using this setup, it effectively capture cases such as object changing sizes as Newton's law violation.

*Secondly*, WorldModelBench is paired with large-scale human labels. We conduct a large scale human annotation procedure and collect **67K** human labels to accurately reflect the performance of existing models with the proposed metrics (§ 3.3). Using these human annotations, we offer several key insights of current video generation models, e.g. insufficient tun-

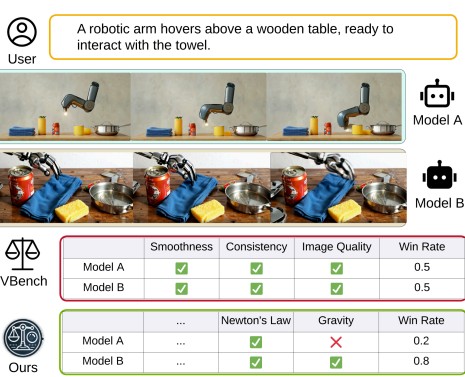

Figure 1: Model A and B generate high quality videos, but the robotic arm in A's video is on the air, violating gravity. Established benchmarks focus on general video quality assessment, and does not distinguish videos that violate physical laws.

ing on I2V models, in §4. We further fine-tune a 2B
parameter judger on the collected human labels to facilitate future model evaluations. We find that the
fine-tuned judger, despite lightweight, learns to predict human preference with 9.9% lower error rate
than GPT-4o [2], thanks to our high-quality human labels. More importantly, we find that aligning
the human annotations by maximizing the scores from the fine-tuned judger improves the world
modeling capability of video generation models [65, 44]. Our contributions are:

1. We demonstrate that previous benchmarks are insufficient for video world models, and
   contribute WorldModelBench to measure world modeling capability of video generation
   models on diverse application driven domains.

2. A large scale of 67K human labels for 14 frontier models, for the community to conduct
   further research.

3. An accurate fine-tuned judger. This judger accurately predicts world modeling violations,
   and fine-tuning on its rewards leads to better generation.

## 2 Related Works

**Video generation models** Many diffusion-based video generation models have made major improvement in synthesizing realistic videos [30, 24, 38, 12, 13, 23, 50, 52, 38, 47, 57, 62, 14, 15, 60, 18, 59, 65, 42, 37, 39, 29, 3, 56]. Many of these models synthesized videos based on input text condition, e.g. [12, 13, 23, 50, 52, 59, 65, 42, 29, 37, 39] image condition [5], or both [56, 57, 65, 30]. In this paper, we focus on evaluation of video models with text and image conditions.

**Evaluation of video generation models.** Previous video generation evaluation mainly uses single-number metric such as Frechet Video Distance (FVD) [48] and CLIPSIM [45]. Huang et al. [26] establishes VBench that provides a comprehensive evaluation on video generation models, focusing on general video quality and video-condition consistency. Wu et al. [54] proposes T2VScore with text-video and general video quality criteria. Bansal et al. [4] further proposes to evaluate videos on whether it follows the correct physics rules in a 0 or 1 granularity. They also keep an instruction following category in a 0 or 1 granularity. Our WorldModelBench further improves along the direction with more fine-grained physics scoring and instruction following scoring, incorporating diverse application domains, and also incorporate previous metrics from VBench. He et al. [22] also uses human annotators, but does not focus on physics and instruction following capability. [27] studies the physics adherence of video generation models on 2D simulation.

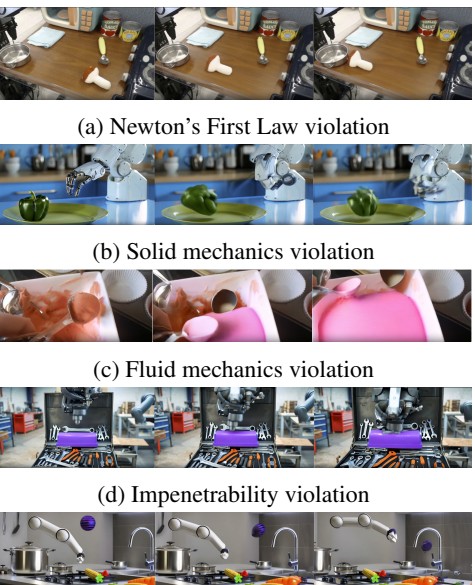

(a) Newton's First Law violation

(b) Solid mechanics violation

(c) Fluid mechanics violation

(d) Impenetrability violation

(e) Gravity violation

Figure 3: Examples of Physics violations.

**Reward models for video generation models** Li et al. [33], Prabhudesai et al. [44] explores using reward models to improve the quality of video generation models. Unlike a rich set of image reward models [58, 55, 28], there is fewer video reward models [33]. VideoPhy collects human labeled data with 0-1 corase labels on whether the model follows instruction or physics. However, they do not further improve the video generation based on the trained reward model. In this paper, we collected a large scale of human preference in video, specifically in the context of world modeling, and train an accurate reward model to reflect human preference.

Learning from reward models has been shown effective to align the model output with human preference in the text domain [32, 43]. In the video generation domain, [61] uses a text-image reward model (RM) to improve the generation quality from human feedback. [33] further extends the idea to use a mixture of text-image and text-video RM to improve model. [44] proposes the reward gradient

framework that incorporates multiple reward models. We follow the reward gradients framework with our fine-tuned judger as the reward model to improve the video generation capability.

# 3 WorldModelBench

In this section, we formally introduce WorldModelBench.

**Design principle** An ideal video world model should synthesize feasible next few frames of the world in response to text (and image) instruction, to facilitate decision-making downstream applications. Thus, the assessment of these models should include: the judgment on the ability to precisely *follow instruction* in input condition, the judgment on the ability to accurately *synthesize next few frames*, and include *diverse application domains*.

Specifically, we breakdown our grading criteria into two parts: (1) **Instruction following**: whether the generated videos correctly follow the text (and image) prompt, and (2) **Future frame generation**: whether the generated videos represents feasible next state of the world, including *physics adherence* and *commonsense*. We introduce fine-grained categories under these two parts in §3.1. The detailed curation procedure is described in §3.2. Finally, we present the procedure for obtaining human annotations in §3.3.

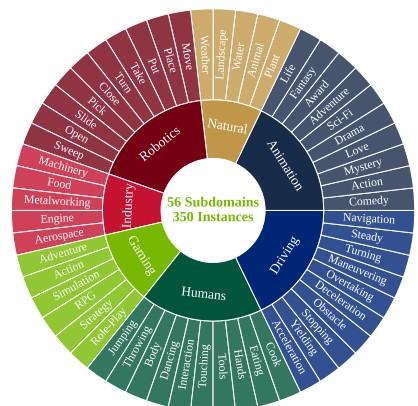

Figure 4: WorldModelBench consists of 7 domains and 56 subdomains, totaling 350 image and text conditions.

## 3.1 Grading Criteria

For each instances in WorldModelBench, a model generates a video based on the text (and image) condition. Each video is then graded in a fine-grained manner along the following dimensions, totaling a score up to 10. Table 1 compares WorldModelBenchwith existing benchmarks.

### 3.1.1 Instruction Following

We define four levels of instruction-following performance and assign scores according to the level (scores 0–3).

**Level 0** The subject is either absent or remains stationary.

**Level 1** The subject moves but fails to follow the intended action. For example, if the prompt instructs a car to turn left, but the generated video shows the car turning right.

**Level 2** The subject partially follows the instruction but fails to complete the task. For instance, if the prompt asks a human to touch their shoulder, but the generated video only shows the human moving their hand toward the shoulder without completing the action.

**Level 3** The subject fully and accurately completes the instructed task.

### 3.1.2 Physics Adherence

Physics laws are the foundational principles of the physical world, and their adherence serves as a critical proxy for assessing the plausibility of generated frames. WorldModelBench evaluates video generation models using five fundamental physical laws, selected based on common failures of contemporary models and findings from related work [4]. Each law is assigned a binary score of 0 or 1, totaling scores from 0 to 5. Examples of violations are illustrated in Figure 3.

**Law 1: Newton's First Law**: Objects does not move without external forces.

**Law 2: Conservation of Mass and Solid Mechanics**: objects do not irregularly deform or distort.

**Law 3: Fluid Mechanics**: Liquid does not flow unnaturally or irregularly.

**Law 4: Impenetrability**: Objects does not unnaturally pass through each other.

**Law 5: Gravitation**: Objects does not violate gravity, such as floating.

### 3.1.3 Commonsense

While measures of general video generation quality is not the main focus of WorldModelBench, they are a prerequisite to a good video world model, i.e., *commonsense*. For instance, a feasible representation of future states needs to have coherent motion and visually reasonable quality. In particular, we follow the categorization of [26], and summarize the commonsense into temporal-level and frame-wise quality. We give a score of 0 or 1 for each quality (total scores 0–2).

**Frame-wise quality**: Whether there is visually unappealing frames or low-quality content.

**Temporal quality**: whether there is noticeable flickering, choppy motion, or abrupt appearance (disappearance) of irrelevant objects.

### 3.2 Curating Procedure for Diverse Domains

WorldModelBench covers a diverse domains of autonomous driving, robotics, human activities, industrial, natural scenes, simulation gaming, and animation. Each domain consists of 50 samples from 5-10 subdomains. Each sample is a text and image condition pair. Figure 4 visualizes the subdomains. To ensure the quality, we perform the following three steps to obtain each sample.

1. **Obtaining a reference video**. To ensure that texts and images condition pairs are feasible, we select a initial sets of videos from existing open license datasets as reference: driving from [11] (CC BY-NC-SA 4.0), robotics from [41] (Apache 2.0) and human activities from [10] (The MIT License). These datasets originally have categories, so we select common ones as our subdomains. We select the reference video of the remaining domains from [40]. Specifically, we use GPT-4o [2] to caption videos and filter keywords of the domains. We also select the most popular subdomains within these domains.

2. **Obtaining the text and image condition.** For each reference video, we select the first frame as an image condition. We use GPT-4o [2] to caption the difference between the first frame and the subsequent frames as the action. We also recaption the image condition to support T2V model. We perform detailed prompt engineering so that the T2V model can have a coherent view of the video (e.g. the objects described in the action will appear in the description of the first frame description).

3. **Human-in-the-loop verification** The previous two steps can introduce errors. For instance, some videos can have black initial frames, the captioning from GPT-4o is not always precise, and some videos do not have potential violations of the grading criteria. Thus, we manually verify all the 350 images and text conditions are of good quality.

### 3.3 Obtaining a Reliable World Modeling Judger

While large (visual) language models have achieved decent agreement with human judgers in domains such as chat assistants [17, 64], it is unclear whether this ability holds true on the world modeling domain, in particular, when it involves subjects such as understanding physics laws. To draw reliable conclusions on contemporary video generation models, we perform a large scale of human annotations. For each vote, we require the human voter to complete a dense annotation with selection of all criteria described in 3.1. In the other words, one complete annotation contains a rich set of 8 human labels on world modeling. Thanks to the scale of our annotations, one generated video can receive more than one vote, which allows us to compute human agreement to validate our vote quality.

Table 1: Comparison of WorldModelBench to other existing video benchmarks: VBench, VideoArena, and VideoPhy.

| | VBench | VideoArena | VideoPhy | Ours |
|---|---|---|---|---|
| *Metrics* | | | | |
| Instruction Following | ✓ | ✗ | ✓ | ✓ |
| Common Sense | ✓ | ✗ | ✗ | ✓ |
| Physics Adherence | ✗ | ✗ | ✓ | ✓ |
| *Support Types* | | | | |
| T2V | ✓ | ✓ | ✓ | ✓ |
| I2V | ✓ | ✓ | ✗ | ✓ |
| *Basic Statistics* | | | | |
| Prompt Suite Size | 946 | 1500 | 688 | 350 |
| Human Label | - | 30k | 73k | 67k |
| Label Release? | - | No | No | Yes |

Table 2: Model performance on WorldModelBench (graded by our judge). Bold and underline indicates the best performance over all models, and open models respectively. "Deform.", "Penetr.", "Grav." is short for "Deformation", "Penetration", "Gravitation".

| Model | Instruction | Common Sense | | Physics Adherence | | | | | Total |
|---|---|---|---|---|---|---|---|---|---|
| | | Frame | Temporal | Newton | Deform. | Fluid | Penetr. | Grav. | |
| Real Videos | 2.97 | 1.0 | 1.0 | 1.0 | 1.0 | 1.0 | 1.0 | 1.0 | 9.97 |
| **Closed Models** | | | | | | | | | |
| KLING [29] | **2.32** | **0.99** | **0.97** | **1.00** | **0.90** | **1.00** | **0.93** | **0.99** | **9.10** |
| Minimax [39] | 2.28 | **0.99** | 0.93 | **1.00** | 0.86 | 0.99 | 0.88 | **0.99** | 8.92 |
| Mochi-official [3] | 2.00 | 0.97 | 0.89 | **1.00** | 0.88 | **1.00** | **0.93** | **0.99** | 8.66 |
| Runway [46] | 2.17 | 0.99 | 0.87 | **1.00** | 0.77 | 0.98 | 0.89 | 0.96 | 8.64 |
| Luma [37] | 1.98 | 0.96 | 0.81 | **1.00** | 0.70 | 0.98 | 0.87 | 0.95 | 8.24 |
| **Open Models** | | | | | | | | | |
| OpenSoraPlan-T2V [30] | 1.72 | 0.83 | 0.85 | 1.00 | 0.77 | 0.99 | 0.91 | 0.98 | 8.04 |
| Mochi [3] | 2.06 | 0.78 | 0.68 | 0.99 | 0.63 | 0.99 | 0.79 | 0.98 | 7.91 |
| CogVideoX-T2V [59] | 2.03 | 0.75 | 0.60 | 0.99 | 0.58 | 0.99 | 0.73 | 0.98 | 7.65 |
| CogVideoX-I2V [59] | 1.78 | 0.61 | 0.52 | 1.00 | 0.52 | 0.99 | 0.68 | 0.99 | 7.08 |
| Pandora [56] | 1.56 | 0.49 | 0.53 | 1.00 | 0.55 | 0.98 | 0.79 | 0.99 | 6.90 |
| T2V-Turbo [34] | 1.37 | 0.64 | 0.44 | 0.99 | 0.41 | 0.99 | 0.73 | 0.98 | 6.56 |
| OpenSora-T2V [65] | 1.61 | 0.40 | 0.29 | 0.98 | 0.30 | 0.98 | 0.64 | 0.97 | 6.17 |
| OpenSora-I2V [65] | 1.42 | 0.36 | 0.18 | 0.98 | 0.22 | 0.98 | 0.68 | 0.98 | 5.82 |

Table 3: The performance of newer models on WorldModelBench, graded by our judge.

| Model | Instruction | Common Sense | | Physics Adherence | | | | | Total |
|---|---|---|---|---|---|---|---|---|---|
| | | Frame | Temporal | Newton | Deform. | Fluid | Penetr. | Grav. | |
| Veo3 | 2.57 | 0.99 | 0.91 | 1.00 | 0.85 | 0.99 | 0.87 | 0.99 | 9.18 |
| Wan 2.1-T2V [49] | 2.30 | 0.99 | 0.96 | 1.00 | 0.86 | 0.99 | 0.95 | 1.0 | 9.04 |
| Wan 2.1-I2V [49] | 2.03 | 0.99 | 0.83 | 1.00 | 0.80 | 0.98 | 0.86 | 1.0 | 8.78 |
| ltx-T2V [21] | 2.38 | 0.93 | 0.83 | 1.00 | 0.80 | 0.98 | 0.86 | 1.0 | 8.78 |
| ltx-I2V [21] | 2.06 | 0.85 | 0.85 | 1.00 | 0.78 | 1.0 | 0.9 | 0.99 | 8.43 |

**Vote statistics** We show the statistics of human votings in Table 4. For basic statistics, we collect 8336 complete votes from student volunteers, translating into 67K labels. We also check the quality of our votes by computing agreement statistics between voters: 87.1% of votes are within an absolute score difference of 2. To inspect the quality of our votes by comparing to related works that are mainly arena-style, we convert our votes into pairwise comparisons. In particular, if a video receives multiple votes, we determine its win or loss against other models on the same prompt by comparing total scores, and report the probability of the same result (win or loss) as the pairwise agreement. We found a 70% pairwise agreement, which is comparable to the $70 \sim 75\%$ in Bansal et al. [4] and 72.8% $\sim 83.1\%$ in Chiang et al. [17]. Furthermore, we select votes from 10 experts that are at least CS PhD level as experts. We compute an interval of 1 standard deviation away from the mean of expert votes. We find that 96.2% and 95.4% of experts and crowd votes fall into this interval, validating the quality from crowd votes.

**Fine-tuning for automatic evaluation** To obtain an automatic judger for future released model, we fine-tune a visual language model(VLM) on the collected annotations [51]. We process a single vote as 8 question answering pair, where the VLM takes in the text (and image) condition and the

Table 4: Vote statistics of WorldModelBench.

| Basic Statistics | | Agreement Statistics | |
|---|---|---|---|
| # complete votes | 8336 | Pairwise agreement | 70.0% |
| # voters | 65 | Score agreement ($\pm 2$) | 87.1% |
| # votes per video | 1.70 | Experts agreement ($\pm \sigma$) | 96.2% |
| # labels | 67K | Crowd agreement ($\pm \sigma$) | 95.4% |

generated videos, and output the score for individual grading criteria in § 3.1. For each prompt, we randomly select 12 generated videos as the training set, and the remaining generated videos as the test set. The results are shown in §4. We found that existing *leading propriety VLM (GPT-4o) achieves decent performance in world model understanding*, providing a new use case for VLM-as-a-judge paradigm. Our fine-tuned judge, with only 2B parameter, efficiently achieves higher accuracy.

Table 5: Model performance on WorldModelBench on human annotations. Bold and underline indicates the best performance over all models, and open models respectively. "Deform.", "Penetr.", "Grav." is short for "Deformation", "Penetration", "Gravitation".

| Model | Instruction | Common Sense | | Physics Adherence | | | | | Total |
|---|---|---|---|---|---|---|---|---|---|
| | | Frame | Temporal | Newton | Mass | Fluid | Penetr. | Grav. | |
| **Closed Models** | | | | | | | | | |
| KLING [29] | **2.36** | **0.94** | **0.92** | 0.93 | **0.88** | 0.96 | 0.89 | 0.93 | **8.82** |
| Minimax [39] | 2.29 | 0.91 | 0.88 | 0.93 | 0.81 | 0.96 | 0.86 | 0.94 | 8.59 |
| Mochi-official [3] | 2.01 | 0.89 | 0.83 | **0.94** | 0.82 | **0.99** | **0.92** | **0.98** | 8.37 |
| Runway [46] | 2.15 | 0.87 | 0.78 | 0.91 | 0.69 | 0.94 | 0.82 | 0.91 | 8.08 |
| Luma [37] | 2.01 | 0.81 | 0.76 | 0.89 | 0.62 | 0.95 | 0.77 | 0.90 | 7.72 |
| **Open Models** | | | | | | | | | |
| Mochi [3] | 2.22 | 0.63 | 0.63 | 0.94 | 0.58 | 0.97 | 0.71 | 0.94 | 7.62 |
| OpenSoraPlan-T2V [30] | 1.79 | 0.70 | 0.77 | 0.9 | 0.66 | 0.97 | 0.89 | 0.93 | 7.61 |
| CogVideoX-T2V [59] | 2.11 | 0.60 | 0.51 | 0.91 | 0.52 | 0.96 | 0.74 | 0.95 | 7.31 |
| CogVideoX-I2V [59] | 1.89 | 0.56 | 0.43 | 0.87 | 0.43 | 0.96 | 0.66 | 0.96 | 6.75 |
| OpenSora-Plan-I2V [30] | 1.77 | 0.47 | 0.54 | 0.84 | 0.42 | 0.97 | 0.70 | 0.92 | 6.62 |
| Pandora [56] | 1.56 | 0.42 | 0.53 | 0.91 | 0.50 | 0.96 | 0.74 | 0.94 | 6.57 |
| T2VTurbo [34] | 1.33 | 0.49 | 0.43 | 0.88 | 0.42 | 0.96 | 0.75 | 0.96 | 6.22 |
| OpenSora-T2V [65] | 1.71 | 0.40 | 0.33 | 0.89 | 0.32 | 0.95 | 0.60 | 0.92 | 6.11 |
| OpenSora-I2V [65] | 1.60 | 0.37 | 0.25 | 0.90 | 0.25 | 0.92 | 0.60 | 0.94 | 5.83 |

## 3.4 Alignment Using the Fine-tuned Judger

VLMs trained on internet-scale visual and text data possess broad world knowledge and strong reasoning capacities, making them promising candidates as "world model teachers". Our judge model, a VLM fine-tuned with human data, is well-suited to provide real-world feedback to enhance video generation models as a more accurate world simulator. We propose a differentiable "learn from feedback" approach to improve a pre-trained video diffusion model using our autoregressive judge.

Building on VADER[44], we formulate our training objectives as follows, given a pre-trained video diffusion model $p_\theta(.)$, an *autoregressive* reward model $R(.)$, a grading criteria $G$, and a context dataset $D_c$. Our training objective is to maximize the reward from the world model judge:

$$J(\theta) = \mathbb{E}_{c \sim D_c, \mathbf{x_0} \sim p_\theta(\mathbf{x_0}|c)} \left[ \sum_{g \sim G} R(\mathbf{x_0}, c, g) \right]$$

(1)

where $\mathbf{x_0}$ represents the generated video. The reward model evaluates the generated video based on key criteria: instruction following, physical adherence, and commonsense as detailed in Section 3, and naively combine all sub-rewards through summation. To address the non-differentiability introduced by the discrete nature of language models, we instead optimize the probability gap of the categorical distribution over the answer tokens (e.g., $p(token("No")) - p(token("Yes"))$, where $p(.)$ represents the cat-

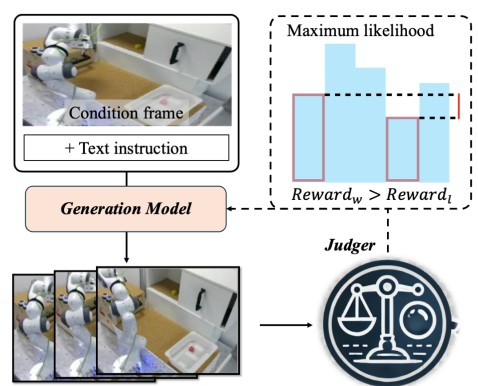

Figure 5: We enhance video generation models by leveraging sparse rewards from our fine-tuned judger. Solid arrows indicate the forward process, while dashed lines are gradient directions.

egorical distribution after softmax for the final hidden states). This method enable us to compute the gradient $\nabla_\theta R(\mathbf{x_0}, c, g)$ and propagate it back to update the parameters of the video generation models.

## 4 Experiments

In the experiment section, we first show and analyze the results of current popular video generation models in our benchmark (§ 4.1) with their absolute average scores, pairwise elo score[17, 16], and per category breakdown scores. Additionally, we follow [17] to demonstrate the quality of the votes being used. Then, we evaluate our fine-tuned judger (§ 4.2), by showing its accuracy in prediction human annotations, and furthermore, the video quality improvement when applying the

reward gradients method with it as the reward model. Lastly, we show ablation studies (§ 4.3) on the scaling effect of number of annotations, and the correlation of our benchmark to the ones in existing VBench [26].

**Models**  We primarily measure 14 models before November 2024. For open-sourced models, we include OpenSora-v1.2 (T2V and I2V) [65], OpenSora-Plan-v1.3 (T2V and I2V) [30], T2VTurbo-v2 [34], CogVideoX-5B (T2V and I2V) [59], Pandora [56], and mochi [3]. For close-sourced models, we include luma-1.6 [37], runway-3.0 [46], minimax [39], kling-v1.5 [29], and an API version of mochi (Mochi-official). We use the recommended hyper-parameters for open-source models (details in the appendix). We also evaluate five addition newer models includeing Veo3, Wan 2.1-T2V, Wan 2.1-I2V, ltx-T2V and ltx-I2v [20, 49, 21] in Table 3.

## 4.1  Evaluation Results

This section analyzes the performance of evaluated models and the quality of the votes.

**Detailed scores**  Table 5 shows scores for all models averaged over all prompts. We present four key observations:

• **Large gap to ideal video world model**: The top scoring model, kling, has only 61% of videos correctly finish the specified task. Furthermore, 12% of the generated videos violate mass conservation law and 11% synthesize objects penetrating each others. This indicates that it not yet has a perfect understanding of properties of physical objects.

• **Better commonsense metrics do not lead to a better video world model.** Luma has higher frame-wise quality (0.81 versus 0.63) and temporal quality (0.76 versus 0.63) scores than the best open model, mochi. Yet, its instruction following capability is much worse than mochi (44% versus 53% videos finish the specified task), and similar physics adherence (4.13 versus 4.14). While previous benchmark [26] mainly focus on the common sense dimension, our results further indicate dimensions that need be considered when training the video generation models.

• **I2V models are worse than their T2V counterpart.** We observe this trend on all three pairs of models (cogvideox 7.31 versus 6.75, opensoraplan 7.62 versus 6.62, opensora 6.11 versus 5.83). This calls for a need to improve the I2V counterpart of released models.

• **Top open models are competitive.** We found that the best open models, mochi and opensoraplan achieve close performance to some closed models (7.62, 7.61 total score versus 7.72 of luma). In particular, mochi has promising instruction following and physics adherence ability.

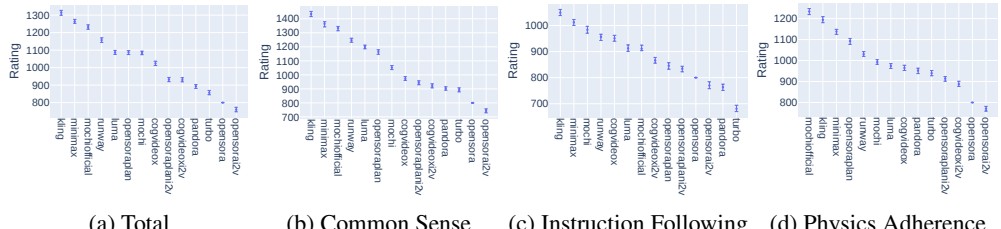

(a) Total  (b) Common Sense  (c) Instruction Following  (d) Physics Adherence

Figure 6: Model ELO rating for categories in WorldModelBench.

**Pairwise comparison**  We further conduct a pairwise comparison of models in Figure 6. We convert our annotations to pairwise setting by enumerating all possible model combination for the same prompt. Following [17], we compute the ELO score using Bradley-Terry model with 100 bootstrapping rounds, using opensora as the 800 ELO calibration. We further observe that there is a **tradeoff** between world modeling capability: e.g. mochi-official has the highest Physics adherence score, yet a middle instruction following score.

**Subdomain breakdown**  We visualize the total scores against all 56 subdomains using heatmap in Figure 7. We find that most models suffer from autonomous driving, human activities and robotics categories, e.g. human throwing objects or jumping. These domains require complex interaction with the environment and accurate modeling of the subject (e.g. human bodies). While most models perform well on natural domains, e.g. on subjects such as plants, animals and water bodies. This calls for a new generation of model that specifically address these hard categories.

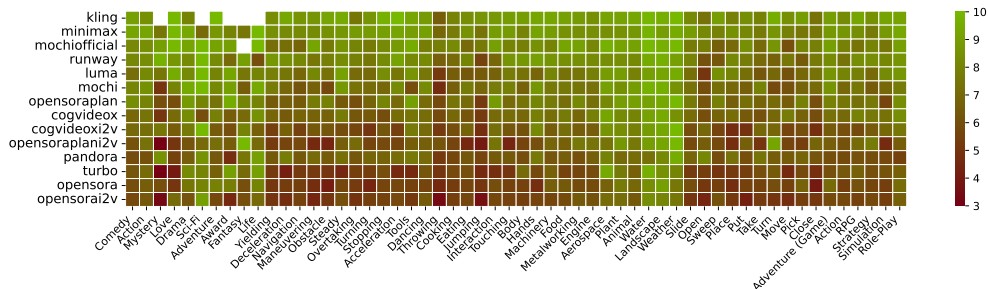

Figure 7: Total scores of model performance visualized with all subdomains. More red colors indicate lower scores; more green colors indicate higher scores. White color denotes missing values due to response refusal from private models.

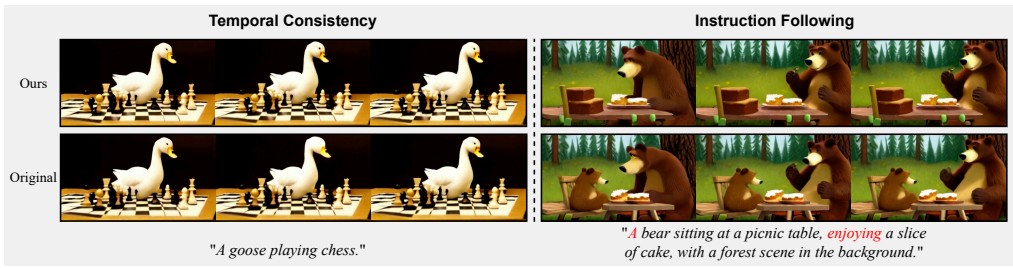

Figure 8: Improvement of our world model gradient method. The bottom row shows videos generated by the original Open-Sora 1.2, while the bottom row features videos produced by the reward-fine-tuned Open-Sora. The original issues of video flickering (left) and instruction non-compliance (right) are mitigated through learning from world model rewards. More results can be found at Figure 11.

## 4.2 Quality of the Fine-tuned Judger

In this section, we show the quality of our fined-tuned judger in two dimensions. Firstly, we compare its accuracy against leading visual language models (GPT-4o) with various strategies on the test set of our benchmark. Then, we show that its score can be used to improve OpenSora-T2V.

**Accuracy on test set** To evaluate the effectiveness of our world model judger, we divide all benchmark votes into a training set and a test set. For each of the 350 prompts, we use videos from 14 different video generation models and annotations from up to 3 distinct voters. We randomly select outputs from 12 models, along with the original video (the video that generates the text prompt and the first frame as conditions, receiving full rewards), to construct the training set, while reserving the rest 2 models for the test set. Our fine-tuned judger is thus trained on a diverse mix of high-reward (high-quality) and low-reward (low-quality) samples, enabling it to effectively distinguish quality differences and predict scores for unseen videos from the same prompts.

Table 6: **Model prediction error results of different judge choices on WorldModelBench**. VILA-2B is a vision-language model with 2B parameters, trained on image and video understanding tasks [35]. We report the average error rate between the model's predictions and the ground truth.

| Model Prediction Error
+*Method* | Instruction (%)
following ↓ | Common (%)
Sense ↓ | Physics (%)
Adherence ↓ |
|---|---|---|---|
| GPT-4o | 29.3 | 35.0 | 36.0 |
| +*CoT* | 29.7 | 28.5 | 45.6 |
| Gemini-1.5-Pro | 30.7 | 34.5 | 29.3 |
| +*CoT* | 29.3 | 19.5 | 28.3 |
| Qwen2-VL-2B | 30.3 | 39.0 | 39.7 |
| VILA-2B +*Zero-Shot* | **21.0** | 28.0 | **24.0** |
| VILA-2B +*CoT Fine-tuned* | 32.3 | **16.4** | 29.7 |

Our dataset includes a total of 4421 videos with 8 human annotations for training, and 713 videos for evaluation (excluding some samples that closed API endpoints refuse). For prompts with multiple votes, we use the majority agreement as the ground truth sparse labels. To enhance alignment with world knowledge and the underlying reasoning processes, we prompt GPT-4o and Gemini-1.5-pro to generate reasoning chains on the training set, and retain chains that reach the correct final answer as additional training data. We then compare our fine-tuned judger's accuracy with different decoding strategies applied to GPT-4o (with zero-shot, and chain-of-thought prompting [53]). Results from

Table 6 show that the find-tuned world model judger achieves higher accuracy than GPT-4o model. We further show comparison between humans and judge scores in Table 10 and Appendix A.4.

**Using the judger as the reward model** We apply the algorithm in § 3.4 with our judger on OpenSora-v1.2 T2V. We show qualitative samples in Figure 8. This shows positive signs for future works to further improve the reward model.

### 4.3 Correlation to Established Benchmarks

Figure 1 provides a motivating example of World-ModelBench, over existing general video quality benchmark. In this section, we conduct an in depth comparative analysis with VBench [25].

We evaluate generated videos on WorldModel-Bench conditions with VBench grading procedure for Opensora, Pandora, Luma, minimax, mochi, Cogvideox, Kling and runway. We compute a pairwise win rate between a pair of models by averaging their pairwise win or loss on the same text (and image) condition, over all available conditions in WorldModelBench, where the win rate $W_{A,B}$ for model $A$ and model $B$ is calculated as follows:

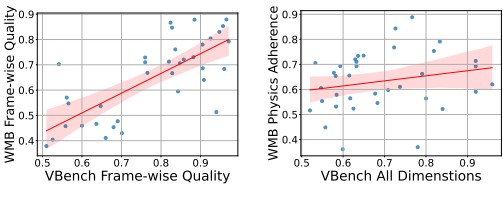

(a) Frame-wise Quality.  (b) Physics adherence.

Figure 9: Correlation between human annotations and automatic metrics.

$$W_{A,B} = \frac{1}{|\text{prompts}|} \sum_{p \in \text{prompts}} \begin{cases} 1 & \text{if eval}_{A,p} > \text{eval}_{B,p} \\ 0 & \text{otherwise} \end{cases}$$

In Figures 9a and 9b, each point represents the win rate between two models, with the x-axis denoting the win rate according to VBench and the y-axis denoting the win rate according to WorldModel-Bench. Figure 9a illustrates the win rates when models are evaluated solely on frame-wise quality, while Figure 9b shows the win rates when models are evaluated based on physics adherence using WorldModelBench and on all dimensions using VBench. We observed a correlation coefficient of **0.69** between the frame-wise quality win rates, indicating a relatively strong correlation. This suggests that both benchmarks are effective in assessing general video quality and that our benchmark aligns with established standards. However, when examining the benchmarks' ability to assess physics adherence, the correlation diminishes significantly to merely **0.28**. This indicates that VBench does not effectively distinguish between videos based on their adherence to physical laws. Supporting this observation, the supplementary material presents an analysis of VBench's other dimension scores, revealing their inability to discriminate based on physics adherence.

## 5 Conclusion

This paper introduces WorldModelBench to evaluate video world models. We found that existing general video quality benchmark is insufficient in evaluating world modeling capability, such as physics adherence. WorldModelBench provides fine-grained world modeling capability feedback to existing video generation models on commonsense, instruction following, and physics adherence dimensions. We collect a large scale of human annotations of 67K to analyze contemporary video generation models as world models. We further fine-tune a VLM to accurately perform automatic judgement on the benchmark. Finally, we show promising signals that maximizing the rewards on the provided judge can improve current video generation models world modeling capability.

## 6 Acknowledgement

We would like to also thank student volunteers from the 6.5940 MIT course (2024 Fall), Qinghao Hu, Guangxuan Xiao, Jiaming Tang, Muyang Li, Shang Yang, Yujun Lin, Zhuoyang Zhang, Haotian Tang, Han Cai, Jinyi Hu, Yuxian Gu, Liuning He from MIT, Enze Xie from Nvidia, Xiuyu Li and Ziming Mao from UC Berkeley, Zeqi Xiao from NTU for helping us to set up annotation pipeline and helpful technical discussions.

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

# A Appendix

## A.1 Correlation to VBench's Dimensions

Section 4.3 illustrates the high correlation (**0.69**) between frame-wise quality win rates of WorldModelBench and VBench, as well as the low correlation (**0.28**) between WorldModelBench's physics adherence win rates and VBench's total score win rates. In this section, we present an analysis of the correlations between WorldModelBench's physics adherence and VBench's other dimension scores.

We compare all VBench dimensions that support customized videos, including subject consistency, background consistency, motion smoothness, dynamic degree, aesthetic quality and imaging quality. Using the same metrics as in Section 4.3, we compute the correlation of model win rates on each VBench dimension and the physics adherence win rates on WorldModelBench. According to Table 7 and Figure 10, the highest correlation coefficient is **0.41** (for aesthetic quality), and the lowest correlation coefficient is **-0.05** (for dynamic degree). Both are significantly lower than the **0.69** correlation coefficient observed for frame-wise quality in Section 4.3. These findings support that VBench does not effectively distinguish videos based on their adherence to physical laws, highlighting the importance of our benchmark in evaluating physical realism.

Table 7: Correlation coefficient of VBench Dimensions with Physics Adherence

| VBench Dimension | Correlation Coefficient |
|---|---|
| Subject Consistency | 0.15 |
| Background Consistency | 0.19 |
| Motion Smoothness | 0.34 |
| Dynamic Degree | -0.05 |
| Aesthetic Quality | 0.41 |
| Imaging Quality | 0.24 |

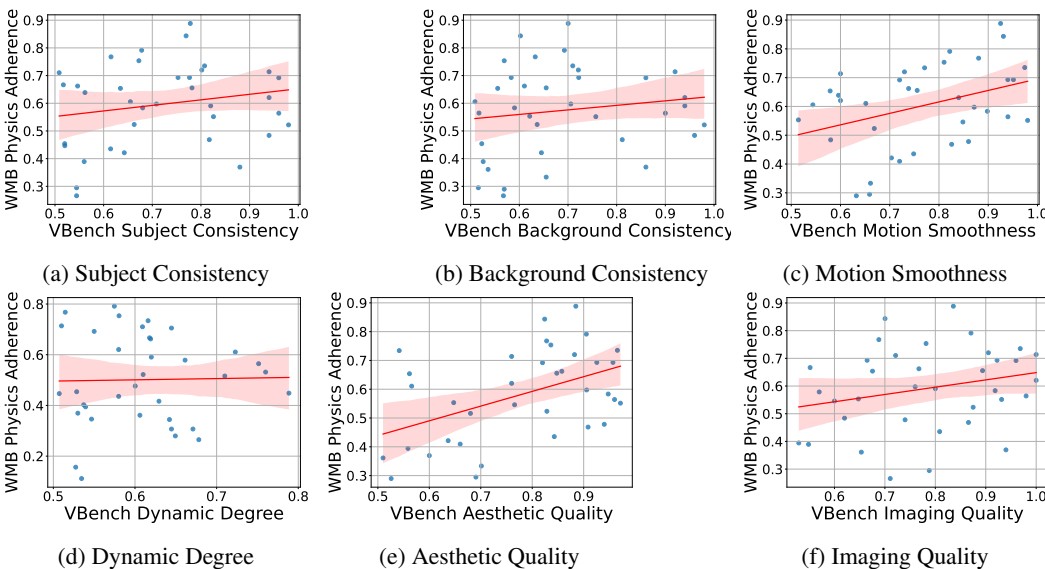

(a) Subject Consistency   (b) Background Consistency   (c) Motion Smoothness

(d) Dynamic Degree   (e) Aesthetic Quality   (f) Imaging Quality

Figure 10: Correlation of model win rates based on all dimensions on VBench and WorldModelBench's physics adherence.

## A.2 More Examples of Reward Optimization

We provide more examples as the results of optimization from the world model judge feedback, as shown in Figure 11. Our method shows potential in leveraging world model feedback to enhance instruction following, improve physics adherence, and achieve better aesthetics, leaving opportunities for future exploration.

## A.3  Model Inference details

We provide the model inference details for open models in our evaluation in section 4.

**CogVideoX [59]** We use CogVideox-5B T2V and I2V model. We use a classifier guidance ratio of 6.0, and 50 step DDIM solver, following the official usage of the model.

**Open-Sora [65]** We use 720P, 4 second, aspect ratio 9:16, 30 sampling steps, with a flow threshold 5.0 and aesthetic threshold 6.5, as recommended by the official website.

**Pandora [56]** We use its official checkpoint, with the default setting provided in the github, with 50 DDIM steps.

**Mochi [3]** we use the default setting with a cfg scale of 4.5, with 65 sampling steps.

**t2v-turbo [34]** We use 4 steps of sampling, 7.5 as classifier free guidance scale, 16 fps and 16 frames as recommended by the official usage.

**Open-Sora-Plan [30]** We use fps 18, guidance scale 7.5, 100 sampling steps, 352 as height and 640 as width as recommended by the official usage.

## A.4  The judge reliability for instruction following

We further demonstrate the judge's instruction following capacity by computing the Kendall rank correlation between the judge predictions and human annotations, and get $\tau = 0.96$ (1 as the max value). We show the score comparison in Table 8, where the average prediction error is 2.79%.

Table 8: **Score comparison between scores provided by humans and by the judge model, on instruction following**. The averaged predicting error is 2.79%.

| Model | Scores ↑ | | Prediction |
|---|---|---|---|
| | **Human (H)** | **Judge (J)** | **Error (100%)** |
| **Closed Models** | | | |
| kling | 2.36 | 2.31 | -2.12% |
| minimax | 2.29 | 2.28 | -0.44% |
| mochi-official | 2.01 | 2.00 | -0.50% |
| runway | 2.15 | 2.17 | 0.93% |
| luma | 2.01 | 1.98 | -1.49% |
| **Open Models** | | | |
| mochi | 2.22 | 2.06 | -7.21% |
| OpenSoraPlan-T2V | 1.79 | 1.72 | -3.91% |
| CogVideoX-T2V | 2.11 | 2.03 | -3.79% |
| CogVideoX-I2V | 1.89 | 1.78 | -5.82% |
| OpenSora-Plan-I2V | 1.77 | 1.76 | -0.56% |
| pandora | 1.56 | 1.56 | 0.00% |
| T2VTurbo | 1.33 | 1.37 | 3.01% |
| OpenSora-T2V | 1.71 | 1.61 | -5.85% |
| OpenSora-I2V | 1.60 | 1.42 | -11.25% |

## A.5  WorldModelBench-Hard

Based on the previous voting results, we curate a smaller hard subset WorldModelBench-Hard to facilitate the model evaluation. Specifically, WorldModelBench-Hard consists of 45 prompts with the lowest average score from the five closed-source models. We provide the detailed score comparison between all models for the hard subset in Table 9. The most performance kling has observed 1.21 regression (from 9.08 to 7.87). These problems are lightweight to evaluate, and also hard enough to distinguish models.

## A.6  Limitations

This section discusses several potential limitations and assumptions in the paper.

Table 9: **Comparison of Judge Model Scores and Hard Subset Scores across Closed and Open Models.**

| Model | Full dataset | Hard Subset Score |
|---|---|---|
| **Closed Models** | | |
| kling | 9.08 | 7.87 |
| minimax | 8.92 | 7.27 |
| mochi-official | 8.66 | 7.24 |
| runway | 8.63 | 7.31 |
| luma | 8.24 | 6.58 |
| **Open Models** | | |
| mochi | 7.91 | 6.93 |
| OpenSoraPlan-T2V | 8.04 | 7.04 |
| CogVideoX-T2V | 7.65 | 6.13 |
| CogVideoX-I2V | 7.08 | 6.27 |
| OpenSora-Plan-I2V | 6.86 | 5.67 |
| pandora | 6.90 | 6.49 |
| T2VTurbo | 6.56 | 5.64 |
| OpenSora-T2V | 6.17 | 4.82 |
| OpenSora-I2V | 5.82 | 4.71 |

Table 10: **Score comparison between scores provided by humans and by the judge model**. The averaged predicting error ($\frac{1}{n}\sum_{i=1}^{n}\frac{Judge-Human}{Human}$) is 4.1%. The highest prediction error is 6.81%, showing the reliablity of our judge model.

| Model | Scores ↑ | | Prediction |
|---|---|---|---|
| | **Human (H)** | **Judge (J)** | **Error (100%)** |
| **Closed Models** | | | |
| kling | 8.82 | 9.08 | 2.95% |
| minimax | 8.59 | 8.92 | 3.84% |
| mochi-official | 8.37 | 8.66 | 3.46% |
| runway | 8.08 | 8.63 | 6.81% |
| luma | 7.72 | 8.24 | 6.74% |
| **Open Models** | | | |
| mochi | 7.62 | 7.91 | 3.81% |
| OpenSoraPlan-T2V | 7.61 | 8.04 | 5.65% |
| CogVideoX-T2V | 7.31 | 7.65 | 4.65% |
| CogVideoX-I2V | 6.75 | 7.08 | 4.89% |
| OpenSora-Plan-I2V | 6.63 | 6.86 | 3.47% |
| pandora | 6.57 | 6.90 | 5.02% |
| T2VTurbo | 6.22 | 6.56 | 5.47% |
| OpenSora-T2V | 6.11 | 6.17 | 0.98% |
| OpenSora-I2V | 5.83 | 5.82 | -0.17% |

**Compare to VideoPhy** VideoPhy focuses on daliy objects, which are not the most relevant domains to world models![4]. We directly measure performance on application domains such as robotics. In addition, WorldModelBench supports image-to-video models, and will open-source fine-grained labels.

**Sample size** WorldModelBench has a considerably a smaller size of other video benchmarks, e.g.,VideoPhy (688). We choose to lower the amount of prompts in our benchmark to enable fast evaluation due to the high inference cost of comtemporary models (e.g. Mochi takes 5 minutes for 4 A100 GPUs). Nevertheless, WorldModelBench is indicative (Table 3): top 2 propriety models has a clear separation (8.82 versus 8.59)

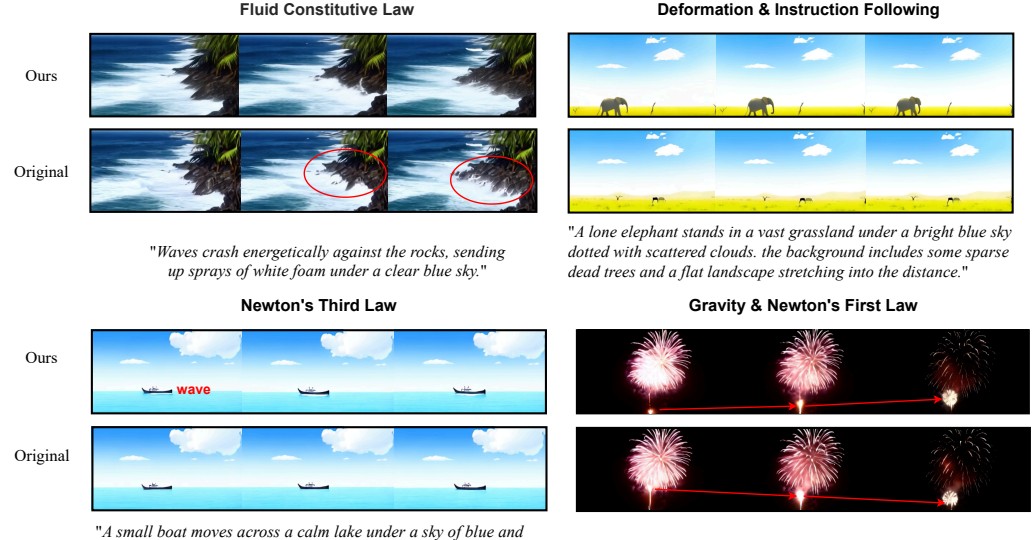

**Fluid Constitutive Law**

Ours

Original

*"Waves crash energetically against the rocks, sending up sprays of white foam under a clear blue sky."*

**Deformation & Instruction Following**

Ours

Original

*"A lone elephant stands in a vast grassland under a bright blue sky dotted with scattered clouds. the background includes some sparse dead trees and a flat landscape stretching into the distance."*

**Newton's Third Law**

Ours

Original

*"A small boat moves across a calm lake under a sky of blue and white clouds, leaving gentle waves rolling backward as it moves."*

**Gravity & Newton's First Law**

Ours

Original

*"Fireworks bloom in the night sky."*

Figure 11: Improvement of our world model gradient method. "Original" shows videos generated by the original Open-Sora 1.2, while "Ours" features videos produced by the reward-fine-tuned Open-Sora. Fine-tuning with the ensembled reward leads to better adherence to world physics, such as: (top left) alleviating the sticky properties of fluids, (top right) recovering from deformation, (bottom left) simulating waves as a result of Newton's third law, and (bottom right) correcting violations of inertia.

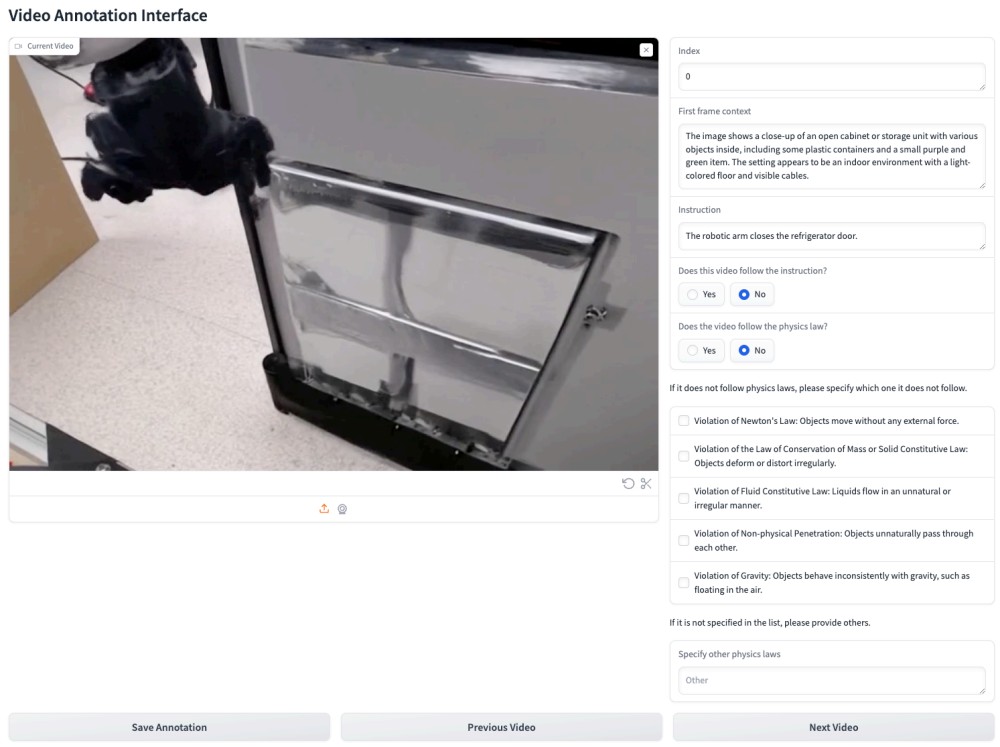

Figure 12: Annotation UI.

