# OpenReview forum: "WorldModelBench: Judging Video Generation Models As World Models"
_NeurIPS.cc/2025/Datasets_and_Benchmarks_Track — NeurIPS 2025 Datasets and Benchmarks Track poster_

### Official Review · Reviewer_KheJ · 2025-07-02

**Rating:** 4
**Confidence:** 4

**Summary:**

This paper introduces WorldModelBench, an evaluation benchmark for video generation models based on world modeling capabilities. WorldModelBench assesses instruction following and physics adherence of the video models. It includes 350 image and text condition pairs across 7 application-driven domains, with 56 subdomains. To establish its reliability, WorldModelBench leverages 67,000 human labels to evaluate 14 frontier models and fine-tunes an automated judger, which outperforms GPT-4o. The authors also demonstrate that aligning training with the judger's rewards can improve the world modeling capabilities of video generation models.

**Dataset Code Accessibility:**

Partly

**Dataset Code Comments:**

The code is not documented, and the evaluation.py in the Huggingface repo cannot be executed with the existing materials in the submission.

**Ethical Considerations:**

No, there are no or only very minor ethics concerns

**Final Justification:**

I appreaciate the authors' feedback with additional experiment on more recent video models and clarification on the code and dataset.
I have read other reviewers' comment and my concerns are addressed. Hence, I'm raising my score to bdl accept.

**Limitations Weaknesses:**

1. The main purpose of the proposed benchmark is to evaluate the world modeling quality of video models, while the existing submission materials do not have any ground truth videos to show the upper bound of these video models for world modeling tasks. The evaluation of ground truth videos should also be included in Table 3 for a thorough comparison.


2. The current submission does not have a well-documented benchmark evaluation code and test set. As detailed in the author guidelines https://neurips.cc/Conferences/2025/CallForDatasetsBenchmarks , the code should be made accessible in an executable format, an empty github repo and a huggingface repo without documentation are not enough to be considered as "accessible in an executable format".



3. In Table 3, more closed-source and open-source methods should be included since all existing ones are published by November 2024, which is far ahead of the submission deadline. For a benchmark paper, it's necessary to keep the leaderboard updated with the current state-of-the-art, e.g., closed-source: Kling-1.6, Veo 2, Seawead-7B, Sora. e.g. open-source: HunyuanVideo, Wan2.1, LTX-Video, Cosmos. Since the proposed benchmark is compared to the previous works in Section 4.3, the authors may refer to the V-Bench leaderboard for their completeness: https://huggingface.co/spaces/Vchitect/VBench_Leaderboard

**Strengths Contributions:**

1. WorldModelBench provides a thorough evaluation of video generation models by incorporating instruction-following and physics-adherence dimensions, which are critical for "world models" but are often missed by previous benchmarks.

2. A large-scale collection of 67,000 human labels across 14 models was used to fine-tune a judge model, ensuring alignment with human preferences. The fine-tuned judger automates the evaluation process with a 9.9% lower error rate than GPT-4o, making future model evaluations efficient.

3. WorldModelBench covers 7 diverse application-driven domains and 56 subdomains, making it relevant for real-world scenarios like robotics and autonomous driving.

---

> ### Author Rebuttal · Authors · 2025-07-31
>
> We sincerely appreciate the reviewer's review and address questions individually below:
>
> Q1: Comparison to ground truth videos “The main purpose … for a thorough comparison”
>
> A1: we evaluate the ground truth videos with our trained judge. We found that our judge grades real videos with a full score on common sense and physics adherence dimension, and almost a full score for instruction following (2.97 out of 3.0) , further validating the quality of our trained judge.
>
> | Model        | Instruction | Frame | Temporal | Newton | Mass | Fluid | Penetr. | Grav. | Total |
> |--------------|-------------|---------------|-------|----------|--------|------|--------|----------|-------|
> | Real videos  | 2.97        | 1.0           | 1.0   | 1.0      | 1.0    | 1.0  | 1.0    | 1.0      |  9.97  |
>
>
> Q2: Code and test set “The current submission … in executable format”
>
> A2: Our code and test set is available in the readme file in the link provided in the manuscript by the submission time :https://github.com/WorldModelBench-Team/WorldModelBench/tree/main?tab=readme-ov-file#evaluation. It was hosted in huggingface as a dataset. To avoid confusion, we have also hosted the evaluation file and data in the github repo directly.
>
> Q3: Newer models after November 2024, “In Table 3… VBench_Leaderboard”
>
> A3: We have added five newer and leading models with API or open-sourced access between November 2024 and now, judged by our judge model. The new results show that the later veo3 has outperformed the original state-of-the-art kling (9.10), where wan2.1 also performs similarly to kling. We also observe similar observation, e.g. the t2v counter-part performs worse than the i2v model, for both wan and ltx models.
>
> | Model        | Instruction | Frame | Temporal | Newton | Mass | Fluid | Penetr. | Grav. | Total |
> |--------------|-------------|---------------|-------|----------|--------|------|--------|----------|-------|
> | Veo3              |    2.57   |  0.99        | 0.91 | 1.0 | 0.85 | 0.99 | 0.87 | 0.99 | 9.18  |
> | Wan 2.1 t2v   |   2.3      | 0.99          | 0.97 |1.0  | 0.89 | 1.0 | 0.89 | 1.0 | 9.04  |
> | Wan 2.1 i2v   |   2.03     | 0.99          | 0.96 |1.0  | 0.86 | 0.99 | 0.95 | 1.0 | 8.78  |
> | ltx t2v             |   2.38    | 0.93          | 0.83 |1.0  | 0.80 | 0.98 | 0.86 | 1.0 | 8.78  |
> | ltx i2v             |   2.06    | 0.85          | 0.85 |1.0  | 0.78 | 1.0 | 0.90 | 0.99 | 8.43  |

---

### Official Review · Reviewer_Pjvn · 2025-07-03

**Rating:** 5
**Confidence:** 4

**Summary:**

This work proposed an automated evaluation framework to quantify the world modelling ability of generative video models. To this end, the authors collect a large dataset of human annotation, which are then used to fine-tune a VLM for automatic judging of categories like physics adherence and instruction following. The authors perform a comprehensive evaluation of current video models, showing the large performance gap to an ideal world model and a discrepancy between common sense metrics and world model metrics.

**Additional Feedback:**

- L8 "Against to nuanced ... violations" missing word?
- L14 judger -> judge (in general, judge would be preferred over judger)
- Fig. 5 the dashed box around the plot makes the dashed arrows hard to distinguish
- Fig. 3 it would be helpful to pair I2V models with their T2V counterpart for easy comparison
- Fig. 6 is too small and very hard to read; the model names do not correspond to Tab. 3

**Dataset Code Accessibility:**

Partly

**Dataset Code Comments:**

The data seems to be complete on HF, but the dataset card is broken.

**Ethical Considerations:**

No, there are no or only very minor ethics concerns

**Final Justification:**

This benchmark addresses an important need for quantifying world modelling capabilities that is not covered by existing benchmarks. The benchmark will undoubtedly be of value to the community.

**Limitations Weaknesses:**

1. There are newer works on benchmarking physics understanding in video models that could also be taken into consideration, e.g., [1]. Specifically for [1], an additional correlation study that could validate the physics adherence scoring of WorldModelBench would be interesting.
2. Models-as-a-judge approaches are hotly debated and commonly suffer from pitfalls like similarity-bias (e.g., [2]). While I believe the authors have done a good job in validating their method and separating models used as a judge from the models that are evaluated, the paper would nevertheless benefit from a discussion of these issues and limitations and how they apply to this specific setup.

[1] Physics IQ Benchmark: Do generative video models understand physical principles? Motamed et al., 2025

[2] Great Models Think Alike and this Undermines AI Oversight, Goel et al., 2025

**Strengths Contributions:**

1. This benchmark addresses an important need for quantifying world modelling capabilities that is not covered by existing benchmarks.
2. The collected human annotations will undoubtedly be of value to the community.
3. The data collection and validation, the training and validation of the judge, and the experiments are comprehensive, methodical, and sensible.
4. The results are interesting and provide (to my knowledge) novel insights into the world modelling capabilities of current frontier models.
5. The paper is well-written with good structure and formatting, which makes it easy to follow.
6. I specifically appreciate the analysis of correlation to existing benchmarks.

---

> ### Author Rebuttal · Authors · 2025-07-31
>
> We sincerely appreciate the reviewer's review and address questions individually below:
>
> Q1: Comparison to other physics understanding benchmark: “There are… interesting”
>
> A1: We believe that validating our score versus other well established benchmarks is valuable. In the manuscript, we cross-validate our frame-wise quality score with the one in VBench (0.7 correlation) [1]. PhysicsIQ [2] studies whether video generation models understand physical principles by e.g., spatial IoU, (MSE) spatial-temporal IoU, MSE to ground truth videos. As a comparison, WorldModelBench uses human annotation to label whether videos adhere to specific laws. Using Gen-3, where both papers measure as an example, we find that it is far from perfect in physics dimension, exemplified by a 0.69 (total achievable 1.0) for Mass Law. Overall, we also found that Gen-3 performs suboptimally compared to other close-sourced models at the submission time (compared to kling, minimax, and mochi-official). On the other hand, while PhysicsIQ uses a different metric, it also finds that Gen-3 is far from perfect (22.8%, total achievable 100%). It also performs suboptimally compared to other close-sourced models (e.g. Figure 6 of [2] shows that Gen-3 performs worse than Lumiere and VideoPoet in Solid Mechanics categories for spatial-temporal ioU). We will add this discussion to the updated manuscript to provide additional insight on how the physics adherence dimension can be compared with ones provided in other works.
>
> [1] VBench: Comprehensive Benchmark Suite for Video Generative Models
>
> [2] Do generative video models understand physical principles?
>
> Q2: Discussion on Models-as-a-judge approach: “Models-as-a-judge … specific setup”
>
> A2: The potential similarity-bias in model-as-a-judge approach has been studied in the context where both the model being judged and the judge are language models [1]. In our context, the model being judged is a video generation model. To the best of our knowledge, there hasn’t been a systematic study on the similarity between a video generation model and a visual language model judge. Nevertheless, we believe this discussion could open up a new direction in the similarity study in video generation with visual language model judgment. We will add this discussion into our manuscript, and leave the similarity study as an exciting future work.
>
> [1] Great Models Think Alike and this Undermines AI Oversight, Goel et al., 2025

---

### Official Review · Reviewer_Hunj · 2025-07-03

**Rating:** 5
**Confidence:** 4

**Summary:**

This paper presents WorldModelBench, a benchmark for evaluating video generation models from a world modeling perspective, focusing on instruction following and physics adherence. It includes 350 text-image prompts, 14 models, and 8,336 full human annotations (≈67K labels). The authors also fine-tune a 2B judger model that outperforms GPT-4o in predicting human preferences, and demonstrate small improvements via reward tuning.

**Additional Feedback:**

Previous work like VideoPhy[1] proposed physics-based evaluation with coarse 0–1 scoring and broad violation types. While the fine-grained rubric in WorldModelBench (0–5 scoring) is a clear extension, it would be helpful if the authors could clarify whether other aspects of the physics evaluation design—such as the selection and formulation of violation types—are newly proposed or directly inherited. This would better highlight the benchmark’s unique contributions and provide clearer guidance for future benchmark design.

[1] Hritik Bansal et al., VideoPhy: Evaluating physical commonsense for video generation.

**Dataset Code Accessibility:**

Yes

**Dataset Code Comments:**

The test set and evaluation scripts areavailable and well-documented. All the materials are sufficient to reproduce benchmark results. Accessibility and clarity are commendable.

**Ethical Comments:**

I do not see major ethics concerns:

- Videos are sourced from permissive‑license datasets (nuScenes, ActivityNet, Open X‑Embodiment, InternVid).

- Crowd annotations follow standard practice; no personal or sensitive data.

- Since the benchmark provides only evaluation prompts and human labels—without involving private data or novel data collection—it poses minimal ethical concerns.

**Ethical Considerations:**

No, there are no or only very minor ethics concerns

**Final Justification:**

I have no further questions now and encourage the authors to include these clarifications in the final version.

**Limitations Weaknesses:**

1. **Lack of prompt-level coverage analysis**: While the benchmark aims to test diverse physical reasoning skills, the paper does not provide a breakdown of how many prompts target specific physical laws (e.g., gravity, collision, penetration). A clearer mapping between prompts and covered physics concepts would help validate its claimed breadth.

2. **Limited failure mode analysis**: Section 4.1 presents high-level observations, and Appendix includes qualitative examples. However, more systematic analysis of where and why models fail—e.g., per-phenomenon breakdown or typical error patterns—could offer valuable insights and guide future model improvements.

3. **Benchmark scale**: With 350 prompts (~ 6 prompts related to each subdomain), the dataset is a solid first step, but may underrepresent certain physical dynamics. Expanding the dataset in future iterations would enhance coverage and robustness.

**Strengths Contributions:**

1. **Well-defined and timely problem scope**：
This benchmark targets a key aspect of video generation: evaluating consistency with physical laws and instruction fidelity. Compared to VBench and VideoPhy, WorldModelBench offers a more fine-grained 10-point rubric—separately scoring instruction (0–3), physics (0–5), and commonsense (0–2)—which helps capture subtle issues like mass conservation violations and object interpenetration. This structured, physically grounded evaluation fills an important gap in existing benchmarks.

2. **Substantial human annotation and judger effort**：
With 8,336 human-annotated video evaluations (67K labels) and strong agreement, the benchmark offers both scale and high-quality supervision for model assessment. It also enables training a 2B reward model that surpasses GPT-4o in aligning with human preference and supports reward-based fine-tuning.

3. **Broad model coverage and accessible insights**：
The benchmark evaluates 14 state-of-the-art models (both open- and closed-source) and presents detailed results through ELO rankings and subdomain heatmaps, offering the community a valuable and interpretable leaderboard for world modeling performance.

---

> ### Author Rebuttal · Authors · 2025-07-31
>
> We sincerely appreciate the reviewer's review and address questions individually below:
>
> Q1: Lack of prompt-level coverage analysis: “While the benchmark … claimed breadth”
>
> A1: We judge all aspects for each prompt to maximize the coverage, i.e., we grade each physics law (Newton, mass, fluid, penetration, gravity) for all prompts.
>
> Q2: Limited failure mode analysis, “Section 4.1 … future model improvements”
>
> A2: We appreciate the reviewer's request for a more detailed analysis of error types. In response, we will include a comprehensive per-phenomenon breakdown in our revised paper. Our analysis reveals that most models often exhibit multiple simultaneous errors. The main recurring error patterns identified are:
> - Identity preservation failure: Objects deform, vanish, or change states across frames, especially in scenes with rapid motion.
> - Physics violations: Objects defy physical laws, e.g., hover without support or phase through barriers.
> - Common sense violations: Scenes/objects exhibit implausible attributes or combinations.
>
> We hypothesize the next-frame objective rewards local token likelihood rather than adherence to conservation laws, the model has no incentive to encode forces, collisions, or gravity. Without explicit physical priors, it freely hallucinates dynam­ics that maximise short-term likelihood but break world knowledge and Newtonian constraints.
>
> Q3: Benchmark scale, “With 350 … robustness”
>
> A3: In the paper, we justified this as a design choice to reduce evaluation cost and show that it offers decent separation for top models. However, we fully value the reviewer’s feedback on adding more prompts and regard it as an important part of our future work.

---

> > ### Comment · Reviewer_Hunj · 2025-08-05
> > **Feedback to authors**
> >
> > Thank you for addressing several of my concerns. First, I have a follow-up question based on the additional feedback I previously provided:
> >
> > Beyond the shift from 0–1 to 0–5 scoring, could you clarify whether the violation types (e.g., gravity, penetration) are newly proposed in this work or adapted from prior benchmarks like VideoPhy? Clarifying this would help better position the novelty of your evaluation design.
> >
> > Also, regarding Q1: thanks for the clarification that all prompts are graded across physical laws. Do you plan to provide summary statistics or categorical breakdowns in the revised version ? (e.g., how many prompts primarily involve gravity, collisions, etc.) ? This would help evaluators better understand the benchmark’s detailed coverage.

---

> > > ### Author Response · Authors · 2025-08-06
> > >
> > > We sincerely appreciate your thoughtful follow-up!
> > >
> > > **On violation types and novelty**: Our overarching evaluation framework for video world models is newly proposed in this work (Section 3). This design principle leads us to evaluate instruction following and future frame generation around physics adherence and commonsense - areas we believe are critical yet under-explored in existing video model evaluations.
> > >
> > > For instruction following, we introduce a new multi-level rubric that captures common failure modes we observed across 14 models (e.g., subject not present, partial motion, incomplete task execution). One related metric in VideoPhy is semantic adherence, yet it only measures a coarse 0-1 video-text alignment without further levels, which we found is insufficient to measure video world models (as subject does not exist is a more severe violation compared to that it does not finish the entire instruction).
> > >
> > > For physics adherence, our categories (e.g., gravity, penetration, mass, etc.) are derived from fundamental physical laws and observations on common failure cases, but our contribution lies in systematically incorporating them into a formal benchmark. While VideoPhy also provides several qualitative failure analyses in the appendix (such as violation of Newton’s second law), it does not use them in the formal benchmark. VideoPhy instead uses a 0-1 score to measure whether the video is physically possible. We found that this is insufficient to measure video world models (as videos that violate more physics laws should be graded lower).
> > >
> > > Additionally, our benchmark focuses on application-centric domains for video world models —such as robotics, autonomous driving, and animation—rather than general physical interactions with daily objects, which further differentiates our work from prior efforts like VideoPhy.
> > >
> > > **On summary statistics**: We greatly appreciate your suggestion regarding categorical breakdowns. In our revised version, we will include detailed summary statistics for each prompt, including the primary physics laws involved. We agree this will provide users and evaluators with a clearer view of coverage.

---

> > > > ### Comment · Reviewer_Hunj · 2025-08-07
> > > >
> > > > Thank the authors for their further clarification. I have no further questions now and encourage the authors to include these clarifications in the final version.

---

### Official Review · Reviewer_jGNx · 2025-07-21

**Rating:** 5
**Confidence:** 3

**Summary:**

This paper introduces WorldModelBench, a new benchmark for evaluating video generation models from the perspective of world modeling with their ability to follow instructions, obey physics, and exhibit commonsense behavior. The authors argue that existing evaluations mostly focus on visual quality (like coherence or realism), overlooking whether videos make logical sense or follow physical laws. To address this, the authors design a benchmark covering seven domains (e.g., robotics, autonomous driving, gaming), with 350 image-text prompt pairs. They gather a large-scale human annotation dataset (67k comparisons across 14 video models) and train a lightweight evaluator model called the Judger, which achieves better performance than GPT-4o on identifying world modeling failures. They also show that using Judger rewards can improve video generation models (e.g., OpenSora) by reinforcing world-consistent behavior.

**Dataset Code Accessibility:**

Yes

**Ethical Considerations:**

No, there are no or only very minor ethics concerns

**Limitations Weaknesses:**

1. Prompt Set is Relatively Small: The benchmark includes 350 prompts, fewer than some prior works like VideoPhy. While the authors justify this as a trade-off to reduce evaluation cost, the smaller size might limit coverage and reduce statistical power across the benchmark’s 56 subcategories.

2. Simplified Physics Scope: The physics checks are based on a small set of canonical laws. While useful, they don’t capture more complex phenomena (like friction, deformation, or fluid dynamics), which could become important as models advance.

3. Subjectivity in Human Judgments: Even with a 70% agreement rate, human preference data can be noisy—especially for nuanced failures.

**Strengths Contributions:**

1. Strong Human Annotation Effort: The paper is backed by an impressive 67,000 human preference labels. The plan to release this data is especially valuable, as it can support future work in model alignment, reward modeling, and evaluation research.

2. A Practical and Competitive Evaluator: The Judger model is compact (2B parameters) yet effective, outperforming GPT-4o on this specialized task. This makes the evaluation pipeline more accessible and less resource-intensive for the community.

3. Demonstrated Impact on Model Improvement: The paper doesn’t stop at evaluation—it also shows that optimizing models with Judger rewards actually leads to better performance, reducing flickering and improving instruction compliance. This gives the benchmark real utility beyond just analysis.

4. Comprehensive Evaluation Across Models: The comparison of 14 prominent models (both open- and closed-source) offers valuable insights into current model limitations. Notably, it highlights that better video quality doesn’t necessarily mean better world modeling, and that text-to-video models generally outperform image-to-video ones in this regard.

---

> ### Author Rebuttal · Authors · 2025-07-31
>
> We sincerely appreciate the reviewer's review and address questions individually below:
>
> Q1: Prompt set is relatively small: “Prompt Set … 56 subcategories.”
>
> A1: In the paper, we justified this as a design choice to reduce evaluation cost and show it empirically provides separation between top models. We fully value reviewer’s feedback that more prompts could improve coverage when we have more subcategories. We regard adding more subcategories and prompts per subcategories as an exciting future work.
>
> Q2: Simplified Physics Scope: “Simplified Physics … as models advance.”
>
> A2: In the paper, we took a step forward from VBench [1] to evaluate physics adherence, and selected violation of Newton, Mass, Fluid, Penetration and gravity laws as the set of criteria based on our observation of commonly violated laws by contemporary video generation models. We fully value the reviewer’s feedback on more complex physics phenomena such as friction could further improve separation between models in the future, and regard adding these aspects as an exciting future work.
>
> Q3: Subjectivity in Human Judgements: “Subjectivity in Human … nuanced failures.”
>
> A3: In this paper, we mainly consider dimensions such as instruction following and physics adherence, which are challenging to traditional metric such as FVD. Thus, we selected to use human preference. In the paper, we use a cross comparison to the number provided in Chatbot Arena (Table 3) as a reference to showcase our voting quality, where their agreement rate is usually 70%+. We fully value the review’s feedback, and regard further selecting a subset from the votes in the work. We will also open-source our voting for the community to conduct research on related subjects.
>
> [1] VBench: Comprehensive Benchmark Suite for Video Generative Models
>
> [2] Chatbot Arena: An Open Platform for Evaluating LLMs by Human Preference

---

### Note · Authors · 2025-08-12

We sincerely appreciate the reviewers’ thoughtful feedback. Reviewers consistently recognized the strengths of our submission:
- Timely, well-scoped focus on evaluating world-modeling capabilities (hunj, Pjvn, KheJ)
- Large-scale human annotation with clear community value (jGNx, hunj, Pjvn, KheJ)
- A practical, competitive evaluator aligned with human judgments that enables efficient automation (jGNx, hunj, KheJ)
- Broad model coverage enabling accessible cross-model comparisons (jGNx, hunj, KheJ)

During the rebuttal, we strengthened the manuscript by:
- Adding targeted failure-case analyses and insights.
- Situating our work against related benchmarks (e.g., VideoPhy) and clarifying novelty.
- Demonstrating the originality of WorldModelBench’s comprehensive evaluation criteria and the reliability of our judge (near-perfect performance on ground-truth videos).
- Reporting evaluations on 5 newer leading models: Veo 3; Wan 2.1 (text-to-video and image-to-video); LTX (text-to-video and image-to-video).

WorldModelBench provides fine-grained, human-aligned, and fully automated feedback on three core dimensions: commonsense, instruction following, and physics adherence, aiming to close a critical gap in assessing real-world dynamics for video generation models. We will open-source all human annotations, code, prompts, and evaluation scripts, and we will incorporate the constructive discussions in this submission. We hope this benchmark advance the evaluation and development of video foundation models in broader application-driven scenarios.

---

### Decision · Program_Chairs · 2025-09-18

**Decision:**

Accept (poster)

**Comment:**

This paper introduces WorldModelBench, a benchmark for evaluating video generation models through the lens of world modeling—specifically instruction-following, physics adherence, and commonsense consistency. The reviewers are broadly positive. I think this paper addresses an important gap in video model evaluation and provides a valuable resource with both methodological novelty and practical impact. While there is room for expansion and refinement, the strengths clearly outweigh the weaknesses. Therefore, I recommend acceptance.

===== FINAL UPDATE FROM DB Track PCs ====

The final decision for this paper has been taken by the program chairs after consultation with the SACs. All Senior Area Chairs have ranked papers according to the feedback from the AC during the review process. We decided to leave the original meta-review to reflect the opinion of the AC in light of the initial discussions with reviewers and SAC.